# Metabolic activity organizes olfactory representations

Wesley W Qian[1,2], Jennifer N Wei[2], Benjamin Sanchez-Lengeling[2], Brian K Lee[2], Yunan Luo[3], Marnix Vlot[4], Koen Dechering[4], Jian Peng[3], Richard C Gerkin[1,2]\*, Alexander B Wiltschko[1,2]\*

[1]Osmo, Cambridge, United States; [2]Google Research, Brain Team, Cambridge, United States; [3]Department of Computer Science, University of Illinois, Urbana, United States; [4]TropIQ Health Sciences, Nijmegen, Netherlands

**Abstract** Hearing and vision sensory systems are tuned to the natural statistics of acoustic and electromagnetic energy on earth and are evolved to be sensitive in ethologically relevant ranges. But what are the natural statistics of *odors*, and how do olfactory systems exploit them? Dissecting an accurate machine learning model (Lee et al., 2022) for human odor perception, we find a computable representation for odor at the molecular level that can predict the odor-evoked receptor, neural, and behavioral responses of nearly all terrestrial organisms studied in olfactory neuroscience. Using this olfactory representation (principal odor map [POM]), we find that odorous compounds with similar POM representations are more likely to co-occur within a substance and be metabolically closely related; metabolic reaction sequences (Caspi et al., 2014) also follow smooth paths in POM despite large jumps in molecular structure. Just as the brain's visual representations have evolved around the natural statistics of light and shapes, the natural statistics of metabolism appear to shape the brain's representation of the olfactory world.

## Editor's evaluation

This computational study provides fundamental insights into the relationship between odors, demonstrating that perceptual similarity is related to proximity in metabolism. The authors use a compelling machine-learning analysis trained on human datasets, which turns out to generalize well across diverse species. The work will be of particular interest to olfactory neuroscientists and researchers looking at sensory representations.

\*For correspondence:
rick@osmo.ai (RCG);
alex@osmo.ai (ABW)

## Introduction

Sensory neuroscience depends on quantitative maps of the sensory world. Color mixing principles (*Young, 1802*; *Smith and Guild, 1932*; *Hering, 1892*) and corresponding biological mechanisms (*Wiesel and Hubel, 1966*; *De Valois et al., 1958*; *Svaetichin, 1956*) help explain the organization of color perception in the early visual system. Gabor filters describe the receptive fields of visual cortex (V1) simple cells in later stages of visual processing (*Jones and Palmer, 1987*). They also account for the organization of acoustic energy from high- to low-frequency and explain the tonotopic representation of perceptual tuning in animal hearing (*Wever and Bray, 1930*; *Gabor, 1947*). Understanding these sensory representations is critical for the design and interpretation of experiments that probe the organization of our sensory world. However, a representation and organizational framework for odor has not yet been established. Even though structure-activity relationships in human olfaction have been explored (*Keller et al., 2017*; *Gutiérrez et al., 2018*; *Gerkin, 2021*; *Kowalewski and Ray, 2020*), 'activity cliffs' – seemingly small changes in molecular structure that produce profound changes

in activity (*Maggiora, 2006*; such as odor) – have limited the generalizability of representations developed from structural motifs (*Gerkin, 2021*). Does such a representation for odor – common to species separated by evolutionary time – even exist?

Machine learning models, particularly neural networks, have identified common representations encoded in biological nervous systems for sensory modalities, including vision and audition (*Yamins and DiCarlo, 2016*). For instance, the first few layers of convolutional neural networks – trained on visual scenes drawn from natural statistics – learn to implement Gabor filters (*Krizhevsky et al., 2012*). More strikingly, learned representations at progressively deeper layers of neural networks predict the responses of neurons in progressively deeper structures in the ventral visual stream (*Zhuang et al., 2021*; *Yamins et al., 2014*). Similarly, neural networks trained to classify odors can also match olfactory system connectivity (*Wang et al., 2021*). Representations of the sensory world learned by training predictive models thus often recapitulate nature (but see *Schaeffer et al., 2022* for counterarguments that warn against overinterpretation). A key idea is that neural networks can build representations of objects that are flexible enough to support transfer learning to new tasks (as shown in *Lee et al., 2022*). In that example, the representational transformation from 'molecule structure' to 'odor label' happens layer by layer; by the penultimate layer (which can be used for the so-called 'embedding' of the representation), most of this transformation has happened. The representation has become odor specific but not necessarily task specific. Thus, that layer can then be used to predict multiple different odor-related tasks.

Here, we perform a comprehensive meta analysis on 16 olfactory neuroscience datasets (*Keller et al., 2017*; *MacWilliam et al., 2018*; *Missbach et al., 2014*; *Hallem and Carlson, 2006*; *Xu et al., 2014*; *Carey et al., 2010*; *Oliferenko et al., 2013*; *Dravnieks, 1982*; *Gupta et al., 2021*; *Pashkovski et al., 2020*; *Del Mármol et al., 2021*), spanning multiple species and levels of neural processing. We find that the embedding from a graph neural network (GNN) trained on human olfactory perception (*Lee et al., 2022*; *Sanchez-Lengeling et al., 2019*), which we term the principal odor map (POM), is highly predictive of the olfactory responses in nearly all datasets, even for species separated by hundreds of millions of years in evolution. In addition, we find POM is specific to olfaction as it shows no advantage in enteric chemoreception tasks or the prediction of general physico-chemical properties. The existence and specificity of POM not only suggest a shared representation of odor across animals but also provide an accurate and computable framework to study the organization of odor space. We show that metabolic reactions that determine the states of all living things – and the odors they emit – explain the organization of POM, and that multi-step reaction paths are smooth trajectories in POM. Finally, we identify strong associations between POM and the natural co-occurrence of molecules in natural substances. Together, these results suggest that the natural statistics of biologically produced molecules shaped the convergent evolution of animal olfactory systems and representations despite significant differences in biological implementation.

## Results
### Neural network embedding as a principal map for animal olfaction

GNNs show state-of-the-art ability to accurately predict human olfactory perceptual labels in both retrospective (*Sanchez-Lengeling et al., 2019*) and prospective settings (*Lee et al., 2022*). Here, we use a GNN embedding – the neural network layer immediately preceding the task-specific architecture – as a representation of odor and evaluate its predictive power in a meta analysis across 16 datasets (Methods) in olfactory neuroscience spanning 9 common model species, including mosquito, fruit fly, and mouse, as well as different scales of biology, including olfactory receptor, glomerular response, cortical neuron response, and whole-animal behavior (*Figure 1a*; *Figure 1—figure supplement 1*). We quantify the predictive performances of the GNN embedding on regression or classification tasks for these curated datasets and compare its performance against generic chemical representations often used in the predictive chemoinformatic models. As shown in *Figure 1b*, the embedding predicts receptor, neural, and behavioral olfaction data better than generic chemical representations across species separated by up to 500 M years of evolution and possessing independently evolved olfactory systems. We thus term this embedding the POM.

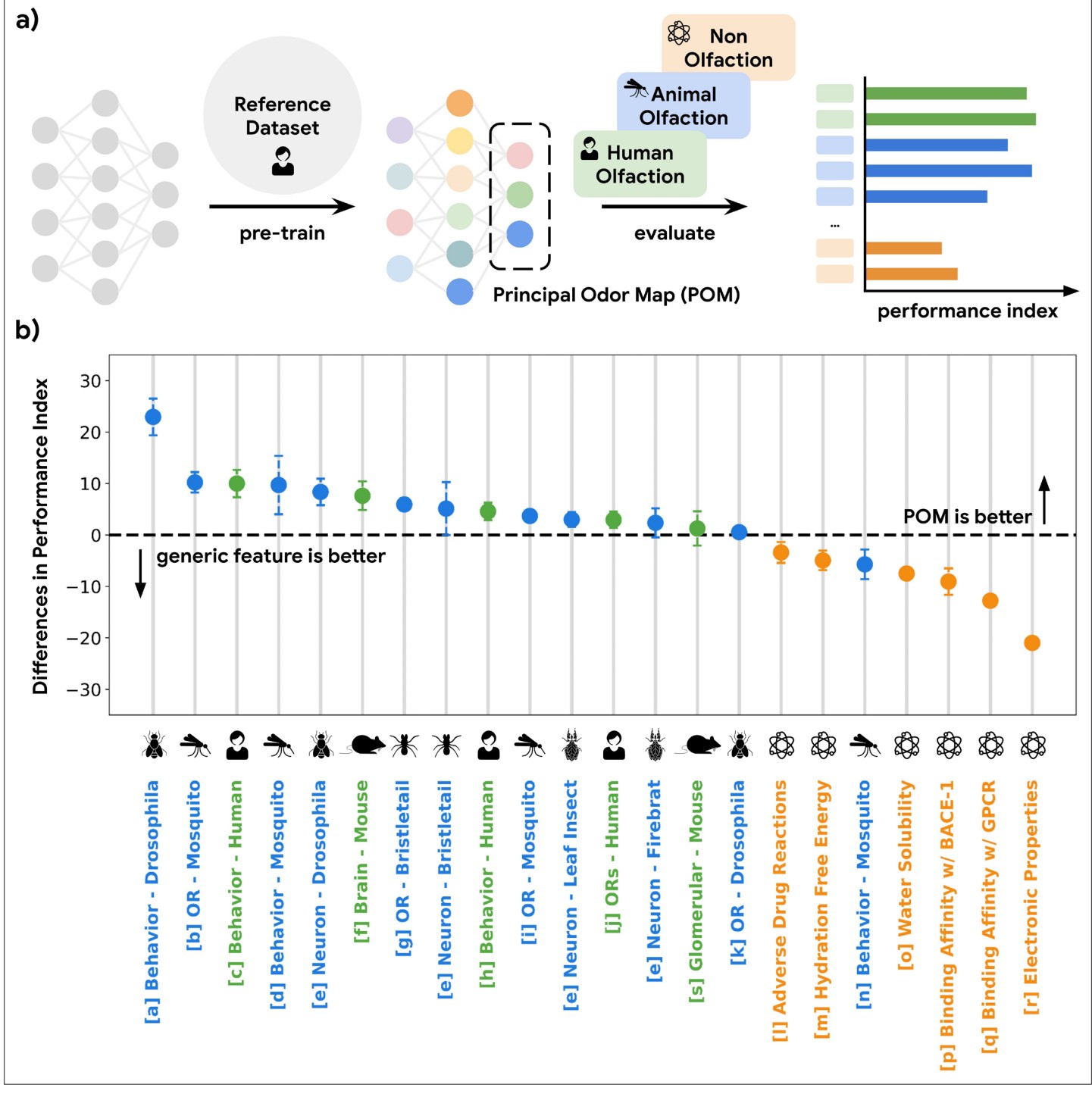

**Figure 1.** A single latent space can explain olfactory data across species and scales. (**a**) A graph neural network model pre-trained on human olfactory perceptual data produces a principal odor map, or POM (latent space, dashed box), which can be used to make predictions about any small, volatile molecule in biological and behavioral experiments. (**b**) A random forest model using only POM produces predictions that meet or exceed those obtained from commonly used generic molecular features (*Moriwaki et al., 2018*; *Bajusz et al., 2015*; Mordred) across a range of olfactory datasets (*Keller et al., 2017*; *MacWilliam et al., 2018*; *Missbach et al., 2014*; *Hallem and Carlson, 2006*; *Xu et al., 2014*; *Carey et al., 2010*; *Oliferenko et al., 2013*; *Dravnieks, 1982*; *Gupta et al., 2021*; *Pashkovski et al., 2020*; *Del Mármol et al., 2021*; *Wu et al., 2018*; *Kuhn et al., 2016*; *Mobley and Guthrie, 2014*; *Delaney, 2004*; *Subramanian et al., 2016*; *Rupp et al., 2012*; *Kooistra et al., 2021*; *Chae et al., 2019*; *Wei et al., 2022*) in different species (green for vertebrates and blue for invertebrates) but not for prediction of non-odorous molecular properties (orange). The Y-axis is the difference between performance indices for models using POM vs. generic molecular features. Performance index is a rescaled metric to place

*Figure 1 continued on next page*

*Figure 1 continued*

classification and regression performance on the same axis. Performance indices of 0 and 100 represent random and perfect predictions, respectively. Error bars are calculated as the SD of performance differences across multiple random seeds.

The online version of this article includes the following figure supplement(s) for figure 1:

**Figure supplement 1.** Performance index for alternative structure-based featurizations.

**Figure supplement 2.** Relative performance of principal odor map (POM) is a function of dataset distance.

**Figure supplement 3.** Principal odor map (POM) has no predictive advantage for non-olfaction-related chemoinformatic properties of odorous molecules.

## The POM is specific for olfaction

While the POM exhibits generalizability across olfactory tasks in various species, it should be no better than generic chemical representations on tasks irrelevant to olfaction in order to optimize its representational power specifically for olfaction (i.e. the no-free-lunch theorems *Roy et al., 2002*). As shown in *Figure 1b*, POM does not show a significant or consistent advantage over generic chemical representations for predicting molecular properties that are not likely exploited by olfaction, such as electronic properties (e.g. QM7 *Rupp et al., 2012*) and adverse drug reactions (e.g. SIDER *Kuhn et al., 2016*) compiled by MoleculeNet (*Wu et al., 2018*). We then apply POM to predict molecular binding activity for G-protein-coupled receptors (GPCRs, of which mammalian olfactory receptors are only a subset *Gupta et al., 2021*) generally, including those involved in enteric chemical sensation (*Kooistra et al., 2021*; e.g. 5HT1A for serotonin and DRD2 for dopamine). While POM demonstrates superior performance for GPCRs involved in human olfaction, their performance is significantly worse for GPCRs related to enteric chemical sensation compared to generic chemical representations, showing specificity for olfaction (*Figure 1b*, *Figure 1—figure supplement 2*). We observe a similar result when we restrict the analysis to only the original training molecules, showing that it is the task and not the molecule which determines the suitability of the POM. (*Figure 1—figure supplement 3*).

## Metabolic activity explains the organization of the POM

Since animals have different biological implementations for external molecular detection (e.g. ionotropic receptors for mosquitoes and independently evolved metabotropic GPCRs for mammals), it is surprising that a human-derived representation of odor can explain responses in a diverse set of species. We hypothesize that such convergent evolution could be the result of a shared natural environment for most animals where they experience the same set of ethological signals, including various nutrients and pheromonal cues from metabolic processes; in other words, detecting and identifying the state of living things by their odor are broadly useful across species.

To test this hypothesis, we explored all odorant molecules in a carefully curated metabolic reaction database called MetaCyc (*Caspi et al., 2014*), containing experimentally elucidated reaction pathways. We identified 17 species with sufficient metabolic data, spanning four kingdoms of life (*Figure 2—figure supplement 1*). We then constructed networks of metabolites for these species in which directed edges represent the direction of a reaction between one node (a reactant) and another (a product; *Figure 2a*). We then computed the discrete 'metabolic distance' between any two compounds by calculating the shortest paths through these networks (*Figure 2b*; *Figure 2—figure supplement 1*). From those metabolic networks with enough metabolites (>100), we repeatedly sampled 50 pairs of odorants (molecules that pass a validated rule set for odor probability; *Mayhew et al., 2022*) for each metabolic distance ranging from 1 to 12 and asked how well the distance in POM correlates with these metabolic distances (*Figure 2c*; *Figure 2—figure supplement 2*). We found that there is a strong correlation between the metabolic distance and POM distance ($r=0.93$) and that common measures of structural similarity between these metabolites can only account for part of the relationship ($r<0.8$). This is especially true for neighbor metabolites in metabolic networks where a biological reaction changes the molecular structure drastically; while such drastic changes produce large structural distances, only smaller changes in POM are observed – the reactant and product spanning this structural cliff frequently share a common odor profile (*Figure 2d*; *Figure 2—figure supplement 3*). Alternative structural distance metrics are also correlated with metabolic distance and with POM distance (*Figure 2—figure supplement 2*); indeed, in the absence of alchemy, metabolic reactants and products must be at least somewhat structurally related. But POM (derived

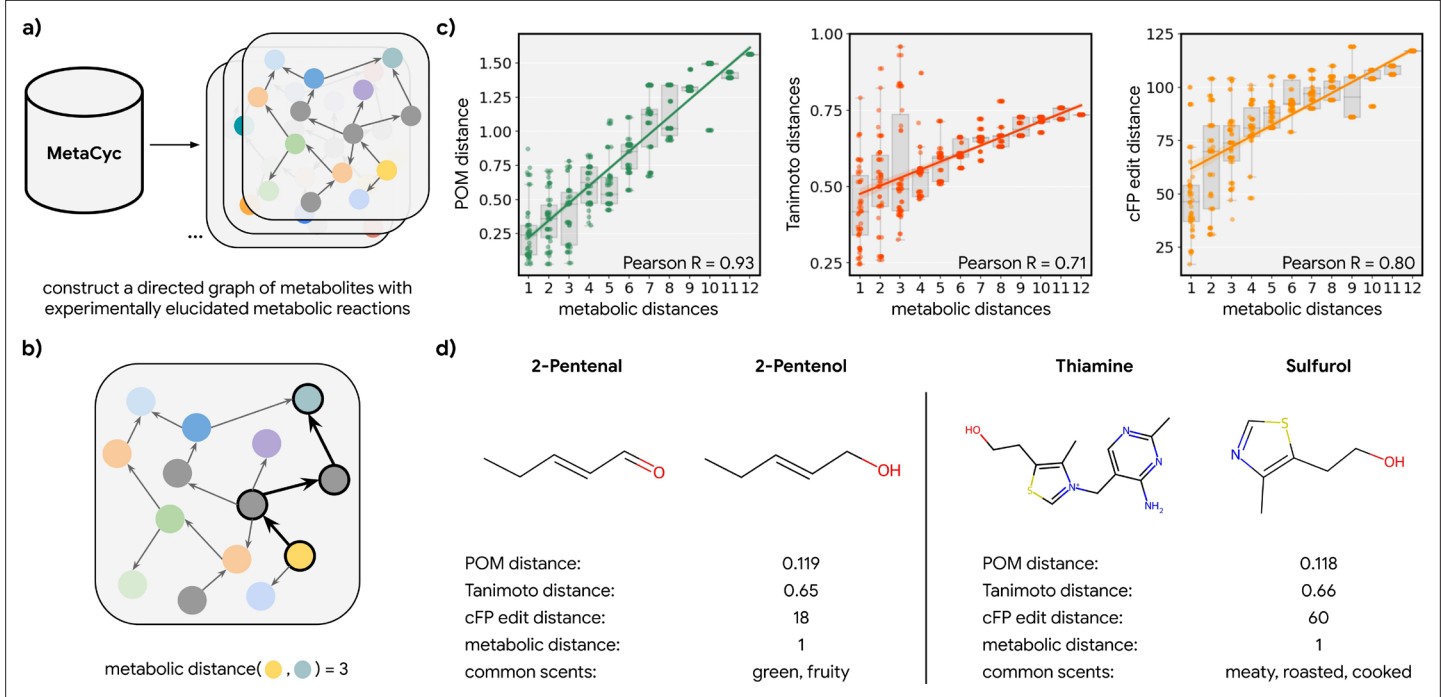

**Figure 2.** Metabolic pathways predict distance in the principal odor map (POM). (**a**) The contents of MetaCyc, a large database of experimentally elucidated metabolic reactions across multiple species, were used to construct directed graphs connecting metabolites, including those with odors (non-gray). (**b**) The discrete pairwise distance of two molecules was defined by the shortest directed path between them within a species' metabolic graph (if any). Each step corresponds to a single chemical reaction specified in MetaCyc. (**c**) Continuous pairwise distances between molecules in POM – which was produced from human perceptual data alone – are strongly correlated with discrete metabolic distance (left, $r$=0.93). This effect is not driven solely by the structure similarity of related metabolites since a weaker relationship is observed using alternative structural distance metrics including Tanimoto distance (center, $r$=0.71) and edit distance in count-based fingerprints (right, $r$=0.80). (**d**) Two pairs of example molecules that are closely related in metabolism. While these are structurally dissimilar molecules (Tanimoto distance >0.65; left: change in a key functional group; right: removal of a major substructure), a single metabolic reaction can turn one to the other, and therefore, POM also organizes them closely together (POM distance <0.12). In turn, they have similar odor profiles.

The online version of this article includes the following figure supplement(s) for figure 2:

**Figure supplement 1.** Distributional statistics for metabolic networks.

**Figure supplement 2.** Correlation between principal odor map (POM) distance and other descriptors.

**Figure supplement 3.** Structural distance vs. principal odor map (POM) distance for one-step metabolite pairs.

**Figure supplement 4.** Correlations are robust to alternative sub-samples of metabolite pairs.

**Figure supplement 5.** Perturbation of the metabolic graph destroys the observed relationships.

only from human perceptual data) outperforms all such metrics significantly, with a >2× reduction in unexplained variance and a lower dispersion at any given metabolic distance (*Figure 2c*; *Figure 2— figure supplement 3*).

Having established that metabolic distance was closely associated with distance in POM, we next asked whether metabolic reactions are easier to understand and interpolate in POM. If a pathway of reactions proceeds in a consistent direction in a molecular representation, then that pathway can be identified with that direction (e.g. 'toward fermentation'); alternatively, the pathway could simply be a random walk in space. Using principal components analysis, we visualized the metabolic pathway for both DIBOA-glucoside biosynthesis (*Figure 3a*) and gibberellin biosynthesis (*Figure 3b*) in 2D with both count-based fingerprints (cFPs) structure and POM. We find the organizations of POM show a smooth progression from starting metabolites to final product metabolites, even though the same pathways show irregular progressions when organized simply by molecular structure. To further quantify such effects, we examine the 'smoothness' of all triplets of three consecutive metabolites (pairs of consecutive reactions) in 37 unique metabolic pathways with only odorant molecules (*Figure 3c*). As shown in *Figure 3d*, most of the paths for these triplets become smoother after the pre-trained neural

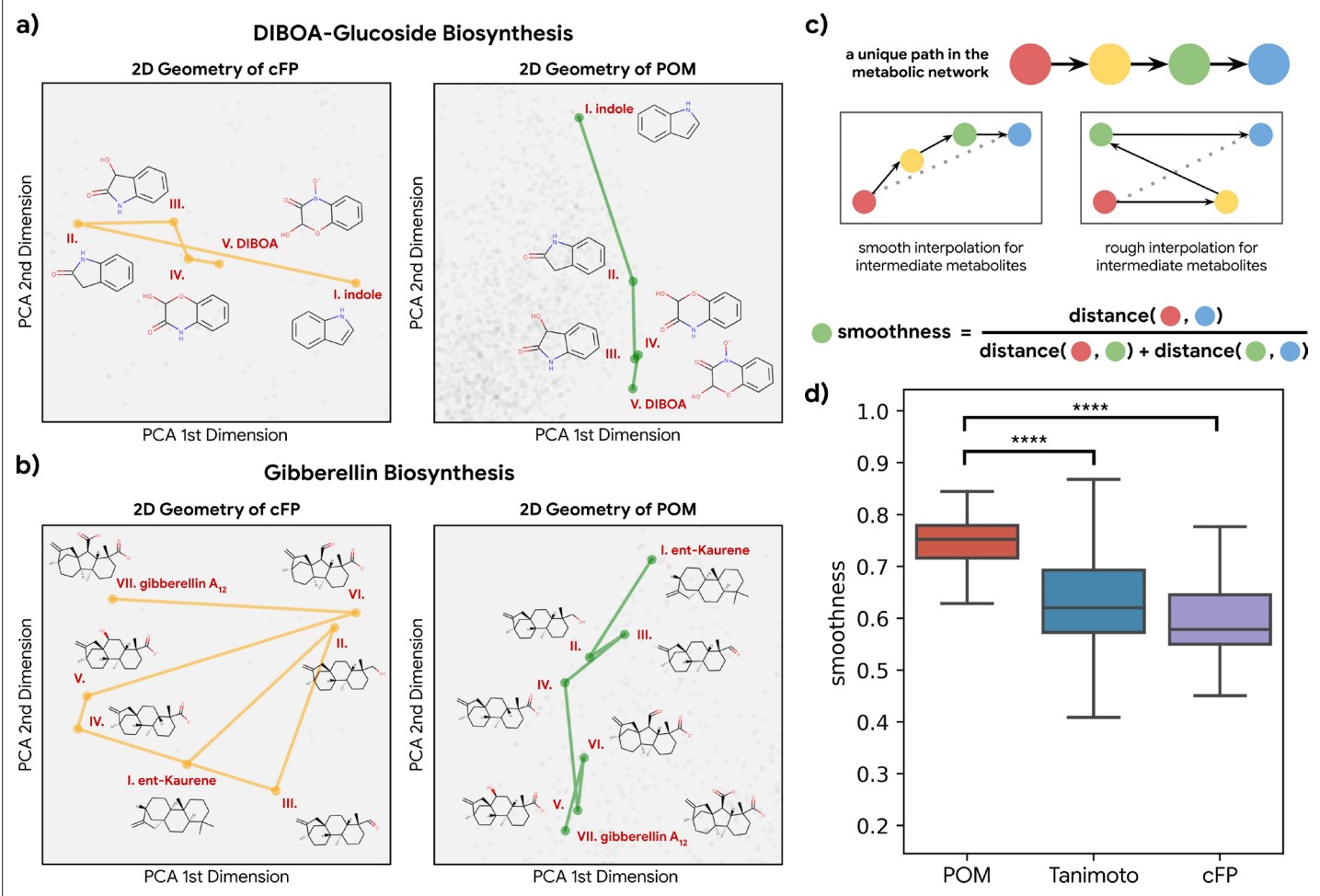

**Figure 3.** Smoothness of metabolic pathways in the principal odor map (POM). (**a**) Left: a four-step pathway (DIBOA-glucoside biosynthesis) depicted in a 2D representation of the structure fingerprints (count-based fingerprints, or cFP) using principal components analysis. Right: the same pathway depicted in a 2D representation of the POM. (**b**) Left: the same 2D cFP representation for a six-step pathway (gibberellin biosynthesis). Right: a 2D representation of the same pathway in the POM. We observe relatively smooth trajectories in POM for these pathways even though the same pathways show irregular trajectories in the structure space. (**c**) To systematically quantify such 'smoothness,' we examine all unique pathways in the metabolic network (top). A desirable molecular representation should exhibit smooth reaction paths, proceeding in a more consistent direction from the starting metabolite to the final metabolite allowing interpolation for intermediate metabolites (center). Smoothness for an intermediate metabolite is formally defined as the ratio between the direct euclidean distance and total path length between the start and end metabolites. A smoother path will result in a ratio close to 1 (bottom). (**d**) Metabolic trajectories are smoother after metabolite structures are projected to POM than when using alternative structural distance metrics (paired t-test, p<0.0001 for both structure distance metrics).

The online version of this article includes the following figure supplement(s) for figure 3:

**Figure supplement 1.** The smoothness of metabolic pathways in principal odor map (POM) is not an artifact of dimensionality.

**Figure supplement 2.** Smoothness of individual metabolite triplets.

network projects their structure into POM (see *Figure 3—figure supplement 1* and *Figure 3—figure supplement 2* for additional controls and analysis), suggesting that the organization of POM reflects a deeper relationship between olfaction and metabolic processes.

## Molecules that co-occur in nature are also closer in the POM

To further validate our hypothesis, we investigated 214 molecules that co-occur in 303 essential oils aggregated in the Pyrfume data repository (*Castro et al., 2022*). Molecules which co-occur in the same object in nature usually convey similar ethological cues, including danger, conspecifics, or in the case of plants, nutrient availability. If the organization of POM is indeed driven by the shared natural

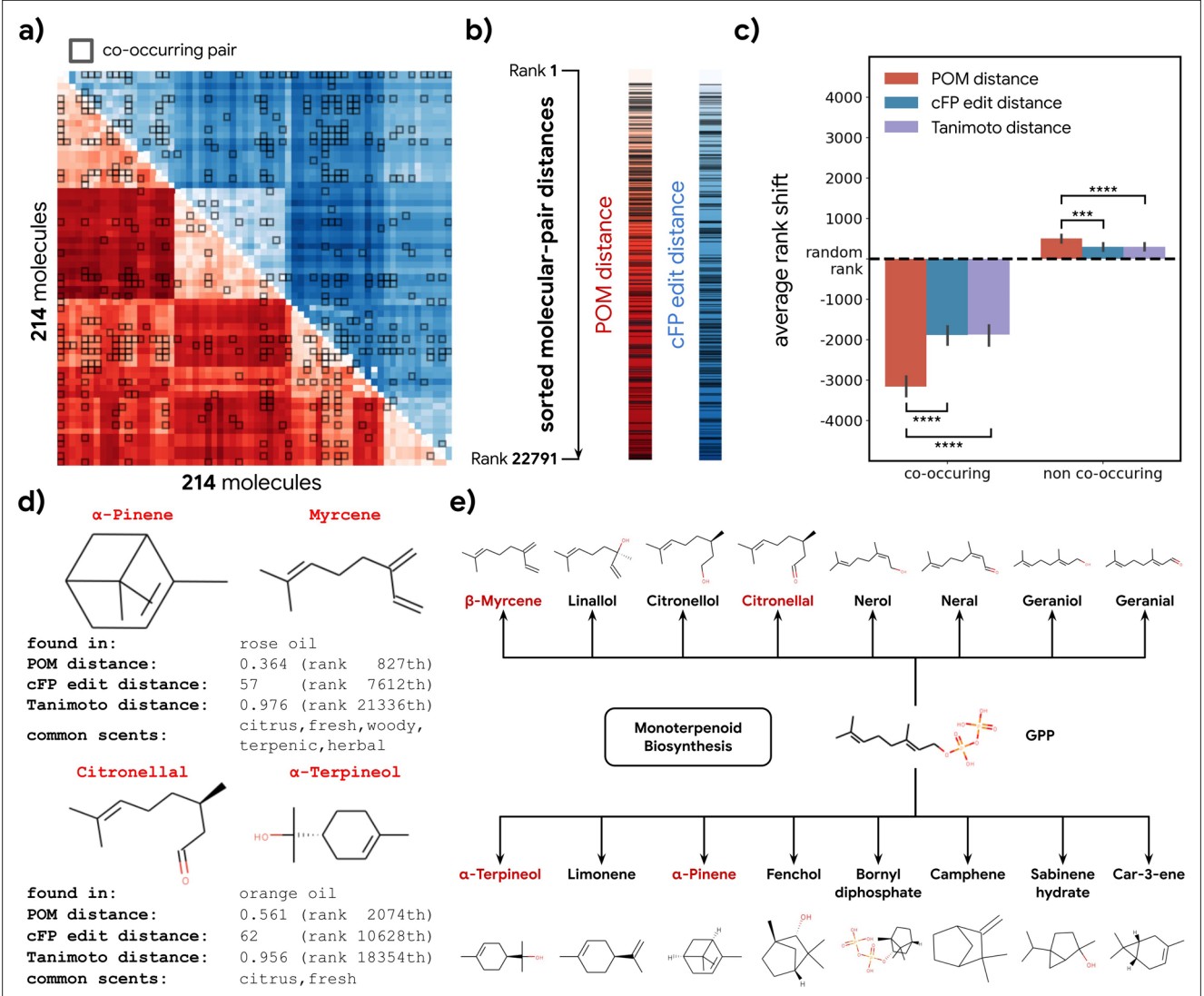

**Figure 4.** Co-occurrence of odorous molecules in natural substances is explained by the principal odor map (POM). (**a**) We compiled a dataset of 214 molecules found in 303 essential oils and computed their pairwise POM distance (red) and count-based fingerprint (cFP) edit distance (blue), where molecule pairs that co-occur in the same essential oil are indicated by dark boxes. (**b**) To make POM and cFP edit distance comparable, we sort both POM and cFP edit distances for all 22,791 molecular pairs in the dataset from small to large and mark co-occurring pairs with dark lines; we then (**c**) plot the average shift in distance rank (relative to a random pair) for co-occurring (left) and non-co-occurring (right) molecule pairs under POM distance (red), cFP edit distance (blue), and Tanimoto distance (purple). As expected, the co-occurring pairs have a smaller rank (nearer together), and non-co-occurring pairs have a higher rank (further apart). More importantly, this rank shift for co-occurring molecules is ~2× larger in POM than for structures distance (paired t-test, p<0.0001) and reversed for non-co-occurring pairs (paired t-test, p<0.001). Error bar indicates the 95% CI. (**d**) Two example pairs of co-occurring molecules that POM successfully recognizes as closely related while conventional structure-based distance fails. Common odor labels for the two molecules as predicted by a state-of-the-art model (*Sanchez-Lengeling et al., 2019*). (**e**) The terpenoid biosynthesis pathway shows that these molecule pairs (red) are close downstream metabolic products of geranyl diphosphate (*Kanehisa et al., 2021*), explaining both their co-occurrence and their proximity in the POM despite their dissimilar structures.

The online version of this article includes the following figure supplement(s) for figure 4:

**Figure supplement 1.** Co-occurring molecules in essential oils also show a smaller neural response distance in insects and mice.

ethological signals driven by metabolic processes, they should also be represented similarly in the POM.

We calculated the distance for all molecule pairs in POM and also using a common structural similarity metric (*Figure 4a*). If co-occurring pairs share metabolic origins, then such co-occurring pairs should be closer together in POM than would be expected by chance. Indeed, the distance

distribution for co-occurring molecule pairs is shifted toward 'nearness' in POM; this shift is larger than one would expect when only considering their structure similarity (*Figure 4b and c*). A similar pattern is observed for animal neural responses (*Figure 4—figure supplement 1*). As an illustration, we sample two pairs of co-occurring molecules – one pair from rose oil and the other from orange oil – that have a very distinct within-pair structure (Tanimoto distance >0.95, *Figure 4d*). Despite their distinct structures, we discover that both pairs of molecules are part of the terpenoid biosynthesis pathway and are closely related downstream products of geranyl diphosphate (*Figure 4e*). The POM representation – in which the distance within each such pair is small – captures their physical co-occurrence and proximity in the metabolome.

## Discussion

In this study, we found that the embedding of a GNN pre-trained on a reference human olfaction dataset can be used as a POM for predicting general receptor, neuron, or animal behavioral olfactory tasks across a large number of datasets. Using POM as an accurate and computable proxy for odor representation, we then proposed a hypothesis for the organization of odor with three facets: (1) co-occurring molecules in nature are also nearby in the POM; (2) the underlying metabolic network dictating co-occurrence is highly correlated with the POM representation; and (3) directed metabolic reaction pathways trace smooth and consistent paths in the same representation. These results suggest that evolution of numerous terrestrial species' olfactory systems has converged to decode a shared and principal set of ethological signals organized by the metabolic processes of nature and the natural statistics that emerge from it. Therefore, we hypothesize that the olfactory *umwelt* may be more similar across species than previously appreciated.

Several previous efforts have 'mapped' odorant molecules and the percepts they evoke (*Mayhew et al., 2022*; *Stevens and O'Connell, 1996*; *Madany Mamlouk et al., 2003*; *Youngentob et al., 2006*; *Khan et al., 2007*; *Castro et al., 2013*; *Bushdid et al., 2014*; *Tran et al., 2019*; *Zarzo and Stanton, 2006*; *Koulakov et al., 2011*), most of which use dimensionality reduction methods, and which are summarized and categorized in a recent review (*Gerkin, 2021*). Many of these share (in common with our approach) an attempt to embed molecules in a latent space defined by chemical features or odor labels, such that 'nearby' odorants are structurally similar, perceptually similar, or both. Our work extends this effort in at least three ways: our training dataset is ~10× larger than most of the others, we use a GNN approach that allows the resulting space to better capture structure/percept relationships, and we test the utility of the space out-of-sample on a multitude of distinct datasets spanning species, brain areas, and biological principles.

Our claim unlocks new directions for olfactory neuroscience, animal ethology, and metabolomics. First, we predict that olfactory neural representations in most animals should be well explained by POM or future representations based on similar principles. Theory suggests that tuning of neurons (*Zwicker, 2019*; *Bak et al., 2018*) and even plasticity (*Oleszkiewicz et al., 2021*) should reflect the natural statistics of odor; these statistics may be organized by metabolic activity and may have esoteric geometries (*Zhou et al., 2018*). Second, the homology between odor space and metabolic space suggests that animal olfactory behavior may be broadly organized around the detection and discrimination of metabolic markers and even holistic metabolic states (*McGee, 2020*): how ripe is this fruit (*Colantonio et al., 2022*)? How healthy is this mate? How nutritious is this soil? Third, the discovery of novel metabolic reactions and pathways could be informed by results from olfaction itself; molecules which have a similar smell – for reasons otherwise unknown – may be neighbors in an ethologically relevant metabolic network. Mechanistically, olfaction mediated by the trace amine-associated receptors (TAARs; *Borowsky et al., 2001*; *Liberles and Buck, 2006*), which detect metabolically downstream products of essential nutrient amino acids, already shows the signature of a 'metabolism detector.' Our results suggest this may be a feature of olfactory chemosensation more broadly. Over both evolutionary time and individual learning and development, the structure of metabolomes could also provide a substrate enabling smooth, separable manifolds to develop in neural activity space, a likely requirement for invariant object recognition (*DiCarlo et al., 2012*).

Indeed, Chee-Ruiter's work in the Bower lab hypothesized this link over 20 years ago (*Chee-Ruiter, 2000*). They identified a metabolic organization to previously published descriptions of the odor qualities of hundreds of compounds and found simple metabolic explanations for several examples of concentration-dependent changes in odor character; for example: some fruity (good) compounds

smell sulfurous (bad) at higher concentration, and these same compounds are linked to decay processes in organic matter. They also showed that humans exhibit perceptual cross-adaptation to compounds metabolically similar (but not otherwise structurally similar) to cis-jasmone, suggesting that the brain's representation of odor similarity is rooted in metabolic similarity. Their work deserves a close read decades later, and this hypothesis is getting increasing attention (*Yang et al., 2023*).

How universal can the POM be, given that outputs of sensory neurons and downstream behaviors are driven by both evolution and learning? Certainly, individual experiences can modify odor perception and act as an additional organizing force on any olfactory map. But environmental correlations are also learned through experience, and many of these are broadly shared across both individuals and species. For example, direct experience of the co-occurrence of citronellal and alpha-terpineol in an orange peel should reinforce, through olfactory plasticity in the brain (e.g. in cortex), those evolutionary changes in the periphery (e.g. receptors) driven by that same co-occurrence. Indeed, we found no clear pattern in relative predictive performance of the POM as a function of processing stage (from periphery to behavior). However, we do observe that the performance of POM improves non-asymptotically as a function of training data size and quality (*Sanchez-Lengeling et al., 2019*); it is thus unclear where the ceiling lies for this approach.

In some networks, connected nodes are not only close in graph distance (e.g. one hop away) but also in physical distance (e.g. same household, same postal code, etc.) Similarly, we show that metabolic distance is closely related to odor distance. But there are surely exceptions; a single edge of a social network graph may span continents, and this exception to the general pattern may be important for explaining the macrostructure of the phenomenon. By analogy, such exceptions for odor, where a single metabolic step gives rise to a radically unrelated odor profile, could represent the boundaries between large, innate odor categories. Given the high correlation between metabolic distance and POM distance that we observe, these are indeed exceptions to a general rule. Future work, relying upon larger metabolic pathway datasets (especially including pathways that have yet to be elucidated), might find enough such exceptions to determine their meaning. Other future experiments could test the hypothesis that correlated odor environments give rise to common representations; for example, animals that are uniquely able to exploit particular food energy sources might also be uniquely adapted to smell metabolic signatures of such sources. In those animals, representations of the corresponding odors might be quite distinct from those found in human perception.

Ludwig Boltzmann and Erwin Schrödinger argued that the fundamental object of struggle for organisms is to identify and feed upon negative entropy (free energy) (*Boltzmann and McGuinness, 1974*; *Schrodinger, 1944*). Each life form appears to be equipped with enzymatic and mechanical tools to access niches of free energy. Eleanor Gibson theorized that perception becomes refined during development to intuitively access this information (*Gibson, 2000*). Indeed, the sense of smell seems designed to identify quantities and accessibility classes of chemical-free energy, which are determined in turn by metabolic processes in living things. Odors thus tend to be more similar within specific pockets of the chemical ecosystem and the carbon cycle. Evolution may have thus ethologically tuned neural representations of chemical stimuli to the statistics and dynamics of this cycle.

## Methods
### POM from pre-trained GNNs

The POM corresponds to the activations of a 256-dimensional embedding layer within a neural network, pre-trained on human olfactory perceptual data (see model architecture and training details below). Briefly, the neural network contains three components: (1) a GNN that represents the molecule as a graph and learns a representation for each atom through message passing, (2) a multi-layer perceptron that aggregates the atom representations and learns an embedding for the entire molecule, and finally (3) a single fully connected layer predicting different odor descriptors. After pre-training the neural network, the parameters of the models are fixed, and the first and second components are used deterministically to generate the location of arbitrary odor-like molecules within the POM; equivalently, the molecule is represented as a graph and projected to a single vector representing the activations of the neural network's final embedding layer. The representation of any odor-like molecule in the POM is this 256-dimensional floating point array embedding.

The analyses carried out upon it are described per figure, but briefly: in *Figure 1*, it was used as the input for predicting datasets from olfactory neuroscience, i.e., the features in a series of regression of classification problems; in *Figure 2*, we computed distances in this space and compared them to distances in alternative spaces; in *Figure 3*, we computed distances and angles in this same space; and in *Figure 4*, we computed distance in the POM between any components found in any essential oil.

## GNN architecture and training details

We pre-trained the GNN on the same human olfactory perceptual data as described in *Sanchez-Lengeling et al., 2019* and *Lee et al., 2022* with 138 odor labels for ~5000 molecules. The model architecture closely follows the implementation of Message Passing Neural Networks (*Gilmer et al., 2017*). For the message passing layer, we used two layers of edge-conditioned matrix multiplication with 63 hidden units and GRU (Gated Recurrent Units) updates, on top of a 45-dimensional atom feature initialization. For the readout layer, we embedded each atom by collapsing the atom features alongside its connected bond features into a 152-dimensional embedding and generate a graph embedding by summing all of the atom embeddings. For the final prediction, we used four fully connected layers with decreasing layer size from 1045 to 256 before making predictions for each of the 138 odor labels from the activations of the last of those layers. To train the model, we optimized the model parameters against a weighted-cross entropy loss in 150 epochs using Adam with a decaying learning rate starting from 5e-4, with a batch size of 128. These hyper parameters are selected using a bayesian optimization algorithm within 100 trials under fivefold cross validation.

## Performance index for supervised learning

Under the supervised learning setting (using molecular featurization to predict out-of-domain results, see *Figure 1*), the performance index of a dataset for a specific representation (e.g. POM or Mordred *Moriwaki et al., 2018*) is calculated from a random forest model's performance using that specific representation as input features. The supervised learning setting includes some datasets with category labels (classification) and some with real number labels (regression).

For datasets of size N≤200, we split the data in a leave-one-out fashion where we hold out one molecule at a time for evaluation and train with N−1 data points until all molecules have been held out once. For each split, N different seeds are used to initiate the model, and the training data is jackknife resampled. For larger datasets of size N>200, we perform a fivefold cross validation split of the data instead, and for each split, 100 seeds are used to initiate the model and the training data is resampled with replacement.

For datasets with categorical labels, we compute AUROC, while $R^2$ score is computed for datasets with real number labels. To calculate the performance index, we then rescale the AUROC or $R^2$ score for each dataset such that 0 represents random performance and 100 represents perfect performance. Specifically, AUROC can be converted to the performance index with (AUROC − 0.5) * 100, and $R^2$ score can be converted to the performance index by multiplying by 100. Finally, performance indices are averaged across all the seeds and targets (in those cases where the dataset contains multiple targets).

Since the optimal hyper parameters for the model can be different for different representations and datasets, we perform a scan for important hyper parameters for random forest models including number of trees in the forest, different ways to assign weight for each class label, number of features to consider during a split, as well as the minimum number of samples required to split and internal node or construct a leaf node. For Morgan fingerprints (cFP and bFP; *Morgan, 1965*), we include an additional hyper parameter to select for the optimal dimension of the fingerprints. In the end, we report the performance index using the best hyper-parameter choices from the scan for each featurization.

## Performance index for mouse piriform cortical activity dataset

Following the original analysis for the dataset in *Pashkovski et al., 2020*, we used correlation distance as the basis for measuring both neural activity distances and molecular representation (e.g. POM or Dragon; *Mauri et al., 2006*) distances for each molecule pair, after centering the values per-neuron or per-feature dimension. Pearson correlations were then used to measure how well each representation

captured the neural activity distances observed in different experimental conditions (representing various parts of the brain and different sets of probes). The performance index is then calculated by averaging and rescaling these latter correlations across each experimental condition.

## Datasets used in figure 1

Each of the datasets below is indicated with a letter ([x]) corresponding to its position in *Figure 1b*.

## Datasets for human olfaction

### Dravnieks [c]

This dataset (*Dravnieks, 1982*) contains 128 unique molecules with 146 odor descriptor targets, where each molecule has a perceptual rating for each odor descriptor, which we can use for regression labels.

### Keller et al. [h]

This dataset is generated from the data published with the crowd-sourced DREAM Olfaction Prediction Challenge (*Keller et al., 2017*) where we used the 'gold' dilution (i.e. the one used to score the challenge) to generate the average perceptual rating for 369 molecules over 21 odor descriptor targets such as pleasantness, grass, garlic, sweaty, etc.

### Human olfactory receptors [j]

This dataset is compiled from the literature and from databases as part of the OdoriFy effort (*Gupta et al., 2021*), and we use the binary response label for all eight different receptor targets including OR1A1, OR1A2, OR1G1, OR2J2, OR2W1, OR51E1, OR51E2, and OR52D1.

## Datasets for mouse olfaction

### Pashkovski et al. (mouse piriform cortex activity) [f]

The activity for each odorant and neuron pair is computed from the original raw time-series response curve kindly provided by the authors of a mouse piriform cortical activity dataset (*Pashkovski et al., 2020*). For each trial, the response curve is first smoothed by averaging each frame with a moving window of size 5. The baseline mean μ and SD σ are then established using activities from the last 30 frames immediately before the designated odor onset. A response is elicited for the trial if the max response value within 30 frames after the onset is larger than $\mu + 3 * \sigma$. The activity for each odorant and neuron pair is represented as the average elicitation rate across multiple trials.

### Chae et al. (mouse glomerular activity) [s]

The activity for each odorant and neuron pair for this dataset (*Chae et al., 2019*) is computed from the dF/F values provided in the curated Pyrfume entry (*Castro et al., 2022*). Each was median-subtracted, such that zero represented a typical baseline response (signal is sparse), and odor-evoked signals were identified with negative dF/F and odor-evoked signals. The normalized score was computed from these odor-evoked signals divided by their mean-squared-deviation from zero signal.

## Datasets for insect olfaction

A total of 11 insect olfaction datasets are organized from 7 prior works in the literature and 1 previously unpublished data source.

### MacWilliam et al. [a]

*MacWilliam et al., 2018* The authors perform a behavioral experiment with *Drosophila* using the T-maze assay. They measure the attraction and aversion with a preference index between –1 and 1 for around 60 compounds, as shown in Figure 5 of their paper. The wild-type preference index for each compound is extracted, and the dataset is represented as a binary classification task, where a compound is considered positive if it can elicit a strong attraction (>0.25) or a strong aversion (<−0.5).

## Xu et al. [b]

***Xu et al., 2014*** The authors measure the odorant-elicited in vitro electrophysiological current response for CquiOR136 in the southern house mosquito, *Culex quinquefasciatus*. The authors measured this response for around 200 compounds as listed in experimental procedures. The compounds that can elicit detectable currents are found in *Figure 3* of that paper and are therefore assigned a positive label in this binary prediction task.

## Wei et al.[d]

***Wei et al., 2022*** As part of a mosquito repellency dataset (***Wei et al., 2022***), 38 molecules were selected from a fragrance catalog, and their repellency was tested with a mosquito feeder assay: the fragrance molecules are coated on the membrane of a nano feeder containing 100 µl of blood meal. After feeding for 10 min, the average percentage of (%) unfed mosquitoes is measured over two trials. The repellency is then calculated by normalizing the unfed percentage such that ethanol has a 0% repellency, while the best possible repellency is 100%. This dataset is formulated as a binary classification task, where a compound is assigned a positive label if more than 90% repellency is observed.

## Missbach et al. [e]

***Missbach et al., 2014*** The authors perform single sensillum recordings for various types of olfactory sensory neurons (OSNs) in four different species including wingless bristletail, *Lepismachilis y-signata*, firebrat, *Thermobia domestica*, neopteran leaf insect, *Phyllium siccifolium*, and fruit fly, *Drosophila melanogaster*. The average spike count per second is recorded for a panel of 35 odorant molecules with six different functional groups. The spike count data is compiled for each species from *Figure 3* of the paper and formulated as a classification task where a compound is labeled as positive for a specific OSN target if the average elicited firing rate is 50% higher than the baseline firing rate.

## del Mármol et al. [g]

***Del Mármol et al., 2021*** The authors measure the odorant-elicited response for olfactory receptors MhOR1 and MhOR5 from the jumping bristletail, *Machilis hrabei*. The authors express the respective receptors in HEK cells and perform whole-cell recordings. An activity index is then computed for a panel of odorants based on the $\log(EC_{50})$ and the maximal response. The data from Supplementary Table 4 and 6 of that paper is compiled and formulated as a regression task with multiple targets.

## Carey et al.[i]

***Carey et al., 2010*** The authors express various olfactory receptors for malaria mosquito *Anopheles gambiae* in 'empty neurons' and measure the olfactory receptor neurons (ORNs) responses as a consequence of odorant stimuli. We cast various receptor ORN responses as different regression targets and compile the data from Supplementary Table 2c of the paper.

## Hallem and Carlson [k]

***Hallem and Carlson, 2006*** The authors perform a similar assay as ***Carey et al., 2010*** but with olfactory receptors in *Drosophila*. The data is extracted from Table S2 of the paper and similarly compiled as a regression task with multiple targets.

## Oliferenko et al.[n]

***Oliferenko et al., 2013*** The authors measure the *Aedes aegypti* repellency for about 90 molecules as the minimum effective dosage (MED). Following the same threshold as the original paper, this dataset is casted as a binary classification task where compounds with an observed MED of less than 0.15 µmol/cm$^2$ are considered active.

## Datasets for non-olfaction related tasks

### Other molecular properties [l, m, o, p, and r]

From MoleculeNet (***Wu et al., 2018***), five diverse tasks are selected as non-olfaction related molecular properties, including electronic properties (QM7 ***Rupp et al., 2012***; ***Blum and Reymond,***

*2009*), binding affinity with BACE-1 protein (BACE *Subramanian et al., 2016*), water solubility (ESOL *Delaney, 2004*), hydration free energy (FreeSolv *Mobley and Guthrie, 2014*), and adverse drug reaction (SIDER *Kuhn et al., 2016*). All these tasks contain a single regression label except SIDER which is a multi-label classification task.

## Enteric GPCR binding [q]

Five large human GPCR targets are pulled from GPCRdb (*Kooistra et al., 2021*) including 5GT1A (serotonin receptor 1a), CNR2 (cannabinoid receptor 2), DRD2 (dopamine receptor 2), GHSR (ghrelin receptor), and OPRK (opioid receptor kappa). Since GPCRdb contains the binding affinities collated from multiple sources, we use the average binding score as our regression label for each target.

## Dataset standardization

All olfactory datasets are standardized by removing the following molecules: (1) data points with multiple molecules (i.e. mixtures), (2) molecules with only a single atom, (3) molecules with atoms that are not hydrogen, carbon, nitrogen, oxygen, and sulfur, or (4) molecules with a molecular weight of more than 500 daltons. Nearly all odorant molecules in the raw datasets passed this empirically motivated standardization filter (*Mayhew et al., 2022*) and are kept in the standardized dataset.

## Metabolic networks and metabolic distance

A metabolic network is constructed for each species where each node represents a metabolite and each directed edge connects the reactant and product metabolites in different experimentally elucidated reactions in the MetaCyc database (*Caspi et al., 2014*). Among all metabolic networks for different species, the 17 largest networks each with more than or equal to 100 metabolites are further studied.

For each metabolic network, all metabolites are labeled odorous or not according to mass transport principles established in *Mayhew et al., 2022*. All pairs of odorous metabolites with an existing path in their network are enumerated, and the distance of their shortest path is used as their metabolic distance – the minimum number of metabolic reactions to convert one odorous metabolite to another. Due to the sparsity of these metabolic networks, far more metabolite pairs with short metabolic distances (<3) are found compared to those with long metabolic distances (>8). In order to fairly study the organization of metabolite pairs across various distances, metabolite pairs are resampled such that an even number (=50) of metabolite pairs is sampled for each metabolic distance. The Pearson correlation coefficients shown in *Figure 2* are also consistent across different sampling seeds (*Figure 2—figure supplement 4*) and are destroyed by perturbations that corrupt the metabolic graph (*Figure 2—figure supplement 5*).

## Computing distances

The POM distance between molecules is defined as the correlation distance between their embeddings, which are centered across the population of molecules (e.g. all sampled odorous metabolite pairs or all compounds in essential oil). Tanimoto distance (*Bajusz et al., 2015*) is computed using RDKit (*The RDKit Documentation, 2019*) based on bit-based fingerprints. cFP edit distance between two molecules approximates the absolute difference between their structures and is defined as the L1 distance between their cFPs.

## Visualizing metabolic pathways and calculating their smoothness

In order to visualize the metabolic pathways (*Figure 3a and b*), both the cFP and POM are projected to a compressed 64 dimensional space via principal components analysis. Using subspaces with a common number of dimensions also controls for any dimensionality bias that might be present in these comparisons. Explaining more than 80% of the variance in both representations, the first two

dimensions of the PCA (principal components analysis) projections are then used to visualize the trajectory of these pathways in cFP vs. POM.

To quantify the 'smoothness' of these metabolic pathways (*Figure 3d*), 34 unique metabolic pathways with distance ranging from 3 to 13 are extracted from the 15 metabolic networks. For an example pathway 'A->B->C->D,' we can calculate the smoothness for interpolation of the two intermediate metabolites B and C. The smoothness for interpolating B can be defined as $d(A, D)/(d[A, B]+d[B, D])$, where $d$ is the euclidean distance between the 64 dimensional PCA projection of cFP or POM (*Figure 3c*). In total, 63 such valid odorous metabolite triplets are found and evaluated.

## Common scents for molecule pairs

The common scents for molecule pairs listed in *Figure 2d* and *Figure 4d* are predicted by a state of the art model (*Sanchez-Lengeling et al., 2019*). An odor label is assigned to the molecule if the model predicts a label probability >0.5 for that label. Multiple labels are possible for each molecule.

# Acknowledgements

This work was done in part while the core team was at Google Research.

# Additional information

### Competing interests

Wesley W Qian: WWQ is an employee of Osmoand was an employee of Google. Jennifer N Wei: JNW is an employee of Google. Benjamin Sanchez-Lengeling: BSL is an employee of Google. Brian K Lee: BKL was an employee of Google. Koen Dechering: KJD holds stock in TropIQ Health Sciences B.V., which collected one of the datasets used in Figure 1. Richard C Gerkin: RCG is an employee of Osmo and was an employee of Google. Alexander B Wiltschko: ABW is an an employee of Osmo, which holds patents WO2020163860A1, WO2022104016A1, EP3906559A1, and BR112021015643A2. ABW does not own equity in Google. ABW was an employee of Google. The other authors declare that no competing interests exist.

### Funding

| Funder | Grant reference number | Author |
| --- | --- | --- |
| Google Research | | Wesley W Qian<br>Alexander B Wiltschko<br>Richard C Gerkin<br>Brian K Lee<br>Benjamin Sanchez-Lengeling |

The funders had no role in study design, data collection and interpretation, or the decision to submit the work for publication.

### Author contributions

Wesley W Qian, Conceptualization, Formal analysis, Validation, Investigation, Visualization, Methodology, Writing – original draft, Project administration, Writing – review and editing; Jennifer N Wei, Conceptualization, Formal analysis, Methodology, Writing – review and editing; Benjamin Sanchez-Lengeling, Software, Investigation, Visualization, Methodology; Brian K Lee, Data curation, Software, Visualization, Writing – review and editing; Yunan Luo, Jian Peng, Formal analysis, Writing – review and editing; Marnix Vlot, Data curation, Investigation; Koen Dechering, Data curation, Supervision, Investigation; Richard C Gerkin, Data curation, Software, Formal analysis, Investigation, Visualization, Methodology, Writing – original draft, Writing – review and editing; Alexander B Wiltschko, Conceptualization, Resources, Supervision, Funding acquisition, Investigation, Methodology, Writing – original draft, Project administration, Writing – review and editing

### Author ORCIDs

Wesley W Qian ⓘ http://orcid.org/0000-0003-0726-575X

Benjamin Sanchez-Lengeling http://orcid.org/0000-0002-1116-1745
Yunan Luo http://orcid.org/0000-0001-7728-6412
Koen Dechering http://orcid.org/0000-0002-8124-4634
Richard C Gerkin http://orcid.org/0000-0002-2940-3378

## Decision letter and Author response

Decision letter https://doi.org/10.7554/eLife.82502.sa1
Author response https://doi.org/10.7554/eLife.82502.sa2

## Additional files

### Supplementary files

• MDAR checklist

### Data availability

Data and code to reproduce all figures is at https://github.com/osmoai/publications (copy archived at *Qian and Gerkin, 2023*).

The following previously published datasets were used:

| Author(s) | Year | Dataset title | Dataset URL | Database and Identifier |
|---|---|---|---|---|
| Caspi et al | 2014 | Metacyc Pathways | https://metacyc.org/group?id=:ALL-PATHWAYS&orgid=META | Metacyc Pathways, id=:ALL-PATHWAYS&orgid=META |
| Caspi et al | 2014 | Metacyc Compounds | https://metacyc.org/group?id=:ALL-COMPOUNDS&orgid=META | Metacyc Compounds, id=:ALL-COMPOUNDS&orgid=META |

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
