## [Editor Report]

This computational study provides fundamental insights into the relationship between odors, demonstrating that perceptual similarity is related to proximity in metabolism. The authors use a compelling machine-learning analysis trained on human datasets, which turns out to generalize well across diverse species. The work will be of particular interest to olfactory neuroscientists and researchers looking at sensory representations.

---

## [Decision Letter]

**Decision letter after peer review:**

Thank you for submitting your article "Metabolic activity organizes olfactory representations" for consideration by *eLife*. Your article has been reviewed by 2 peer reviewers, one of whom is a member of our Board of Reviewing Editors, and the evaluation has been overseen by Andrew King as the Senior Editor. The reviewers have opted to remain anonymous.

Essential revisions:

1. The reviewers agreed that the framing of the problem is interesting, and the idea that odor perceptual distance relates to metabolic proximity is worth exploring. Can the authors cite a highly relevant PhD thesis that also discussed this many years ago: Chee-Ruiter (https://thesis.library.caltech.edu/7595/)

2. A comparison of neural data for a range of presented molecules would greatly strengthen the current argument which is mostly based on human perceptual classification.

3. Can the authors more closely compare the POM distance vs the Modred descriptor, especially indicating correlations between the measures?

4. Can the authors provide substantially more detail about the technical details of the GNN and how readers can access the data and run the calculations? The code should be open source.

*Reviewer #1 (Recommendations for the authors):*

Can the authors place their findings in context with numerous prior attempts to come up with an odorant perceptual space, typically using PCA and similar methods? The authors mention some studies very briefly. It would be interesting to understand where the current approach diverges from the findings of earlier work.

The description of the methods is inadequate, especially since this is a general biology readership. The authors should provide considerably more detail on the GNN architecture, and the training methodology, and refer the reader to the source code. If the authors have done an additional level of curation/organization of the data, that too should be made freely accessible. For computational studies, it is preferable to provide user-downloadable scripts to generate key figures.

Can the authors explain some of the theoretical and ANN-field background of the approach to taking a subset of a GNN as an embedding that encapsulates the dataset referred to as POM in this study.

Can the authors spend substantially more time explaining the features of the POM: Its size, number of parameters, and how it is interrogated in the various analyses carried out in this study?

The authors make a very interesting link to metabolism by comparing the odor distances with the metabolic distances. Again, for reproducibility and further work it would be good to provide the database of these extracted metabolic distances.

*Reviewer #2 (Recommendations for the authors):*

1. In much of the paper, it is claimed that models such as neural networks resemble brain representations (eg second paragraph of Intro: "Representations of the sensory world learned by training predictive models thus often recapitulate nature."). Increasingly, there are counterexamples and arguments against the generality of this assertion. The discovery of internal structure in "such models may be more strongly driven by particular, non-fundamental, and post hoc implementation choices than fundamental truths about neural circuits or the loss function(s) they might optimize." (Quoted from https://www.biorxiv.org/content/10.1101/2022.08.07.503109v1)

2. There are additional data sets that might be worth analyzing. Of course, the authors have plenty already, but I wonder why no mouse receptor or glomerular data were used (given that many groups have produced tons of data on this) – just this reviewer's curiosity.

3. For historical reasons, the authors may consider citing a PhD thesis on the relationship between metabolic similarity and perceptual similarity by Chee-Ruiter (https://thesis.library.caltech.edu/7595/). I'm not sure if that work was ever published in a peer-reviewed journal.

[Editors’ note: further revisions were suggested prior to acceptance, as described below.]

Thank you for resubmitting your work entitled "Metabolic activity organizes olfactory representations" for further consideration by *eLife*. Your revised article has been evaluated by Andrew King (Senior Editor) and a Reviewing Editor.

The manuscript has been substantially improved but there are some remaining issues that need to be addressed, as outlined below:

1. Can the authors provide a more complete explanation of the analysis methods for Figure 4 supplement 1? There are very brief accounts in the methods section on pages 15 and 16. How does one go from ORN responses to regression targets and thence to neural distance?

2. The authors may wish to put some of the key points about Mordred vs POM measures from their responses to the reviews, into the text or discussion.

3. Some fixes needed in the text:

Page 12: "…in Figure 4 we computed distance metrics upon it is the distance between two components of an essential oil."

4. Figure 4 supplement 1: Carey et al. plot seems to be missing bars for cFP – edit distance.

5. Figure 4 supplement 1: Chae et al. seem to have a much smaller rank shift. Is this correct? Is it just due to the number of odors in the sample, or does it relate to the difference in the scoring function as described (very briefly) in the text?

6. Figure 4 supplement 1: No stats to compare the bars?

7. Tanimoto distance should be spelled throughout with a capital T.

---

## [Author Response]

Essential revisions:1. The reviewers agreed that the framing of the problem is interesting, and the idea that odor perceptual distance relates to metabolic proximity is worth exploring. Can the authors cite a highly relevant PhD thesis that also discussed this many years ago: Chee-Ruiter (https://thesis.library.caltech.edu/7595/)

This is a fantastic find and we are very appreciative of the reviewer for pointing this out to us. We have dedicated a paragraph in the discussion to this wonderful work.

2. A comparison of neural data for a range of presented molecules would greatly strengthen the current argument which is mostly based on human perceptual classification.

We have added a new figure (Figure 4—figure supplement 1) which uses neural data (from animal neuroscience experiments) to address the same question, and from which we draw the same conclusion.

3. Can the authors more closely compare the POM distance vs the Modred descriptor, especially indicating correlations between the measures?

We have added a new figure (Figure 2—figure supplement 2) which compares these measures by showing their correlation for the data used herein.

4. Can the authors provide substantially more detail about the technical details of the GNN and how readers can access the data and run the calculations? The code should be open source.

We have added additional technical detail about the model and its implementation in the methods. These details should also be sufficient for an experienced ML engineer to train a similar model.

The code for reproducing all figures from the model embeddings is open source, as are the embeddings themselves for thousands of molecules of interest to neuroscience, which is sufficient to verify and extend our findings. This was perhaps obscured by being shared in the data availability statement rather than the Methods. We have added links in the methods to this code and data.

Reviewer #1 (Recommendations for the authors):Can the authors place their findings in context with numerous prior attempts to come up with an odorant perceptual space, typically using PCA and similar methods? The authors mention some studies very briefly. It would be interesting to understand where the current approach diverges from the findings of earlier work.

There have indeed been numerous previous efforts to generate “perceptual space”, including the beautiful PhD thesis mentioned below by Reviewer #2. These include but are not limited to Stevens and O’Connell 1996, Mamlouk et al. 2003; Youngentob et al. 2006, Khan et al. 2007, Zarzo and Stanton 2005, Koulakov et al., 2011, Castro et al., 2013, Bushdid et al., 2014, Tran et al., 2019, Mayhew et al., 2022, most of which use dimensionality reduction methods, and which are summarized and categorized in a recent review (Gerkin, 2021, Chemical Senses). While the full context is best explicated in a review, we have raised a few of these in the discussion, noting key similarities and differences with our approach. The most important key difference is that previous spaces built from perception are typically limited to those molecules which have been perceived (i.e. a few thousand or fewer), and those built from structure so far don’t appear to have captured the structural differences that really matter for perception, differences which we argue may be closely related to the kinds of “edits” made by metabolic reactions.

The description of the methods is inadequate, especially since this is a general biology readership. The authors should provide considerably more detail on the GNN architecture, and the training methodology, and refer the reader to the source code. If the authors have done an additional level of curation/organization of the data, that too should be made freely accessible. For computational studies, it is preferable to provide user-downloadable scripts to generate key figures.

We agree that the computational methods (model architecture) details can be expanded, and we relied too heavily on citation to our previous work in the initial submission. We have added more details about the architecture (adapted in part from our previous work) to the Methods section, and also provided more details about model training.

As for the data and reproduction materials, we did in the original submission provide a link to all of the datasets used to generate the figures and to Jupyter notebooks to reproduce them, given the model embeddings for each molecule, which we also provided. However, we had only provided these in the Data Availability Statement, which may have been opaque. We have since moved this information to the Methods section, and we refer the reviewer to that site again if they are interested in reproducing our work.

Can the authors explain some of the theoretical and ANN-field background of the approach to taking a subset of a GNN as an embedding that encapsulates the dataset referred to as POM in this study.

Yes, this is a common approach for building representations of objects that are flexible enough to support transfer-learning to new tasks (as shown in Lee et al., 2022, “A Principal Odor Map Unifies Diverse Tasks in Human Olfactory Perception”, for example). The key idea is that the representational transformation from “molecule structure” to “odor label” happens layer by layer, and that by the penultimate layer (which we use for the embedding), most of this transformation has happened and the representation is odor-specific but not necessarily task-specific. That layer can then be used to predict multiple different odor-related tasks. See Zhuang et al., 2017 (https://arxiv.org/abs/1911.02685) for additional examples and background. We have revised the Introduction to introduce this idea to the reader.

Can the authors spend substantially more time explaining the features of the POM: Its size, number of parameters, and how it is interrogated in the various analyses carried out in this study?

We have revised the Methods to include this information more explicitly; some of this was included in the original version but perhaps in a form that was not clear to the reader. Briefly, it has 256 dimensions, i.e. each molecule is represented by a length-256 floating point array. The analyses carried out upon it are described per figure, but briefly: In Figure 1 it was used as the input for predicting datasets from olfactory neuroscience, i.e. it was the “X” in a series of regression of classification problems; in Figure 2 we computed distances in this space and compared them to distances in alternative spaces; in Figure 3 we computed distances and angles in this same space; and in Figure 4 we computed distance metrics upon it is the distance between two components of an essential oil.

The authors make a very interesting link to metabolism by comparing the odor distances with the metabolic distances. Again, for reproducibility and further work it would be good to provide the database of these extracted metabolic distances.

In the previous submission (in a URL in the data availability statement) we provided a notebook to reproduce this information from the source data, as part of the analysis steps for reproducing Figure 2. For the revision we have added an explicit table

(metabolite_analysis/data/metabolite_distance.csv) at the Github link (see Methods) containing the metabolic distances and descriptors distance for each pair of metabolites in different species’ metabolic network, which represents an intermediate step in that analysis, and added the GitHub URL to the Methods section to increase visibility.

Reviewer #2 (Recommendations for the authors):1. In much of the paper, it is claimed that models such as neural networks resemble brain representations (eg second paragraph of Intro: "Representations of the sensory world learned by training predictive models thus often recapitulate nature."). Increasingly, there are counterexamples and arguments against the generality of this assertion. The discovery of internal structure in "such models may be more strongly driven by particular, non-fundamental, and post hoc implementation choices than fundamental truths about neural circuits or the loss function(s) they might optimize." (Quoted from https://www.biorxiv.org/content/10.1101/2022.08.07.503109v1)

We have revised the language used in the Introduction to include this caveat and included a citation to the work above.

2. There are additional data sets that might be worth analyzing. Of course, the authors have plenty already, but I wonder why no mouse receptor or glomerular data were used (given that many groups have produced tons of data on this) – just this reviewer's curiosity.

Some datasets are underpowered to test the hypothesis, or collected in a way that makes it difficult. For example, Burton et al. 2022 reports the activity of glomeruli in response to odorants, but they choose concentrations to find the most sensitive glomerulus for each odorant, which is a very valuable activity but leaves us without a classification or regression problem to test. We instead added Chae et al., 2019, which reports glomerular activity more broadly for a range of odorants, and have included the result in the revised Figure 1. Author response image 1 is a detailed performance breakdown across 100 random samplings, using that glomerular activity dataset. Compared to the best baseline descriptor (Mordred), POM still shows a statistically significant (p < 5e-5) improvement.

**Author response image 1. sa2fig1:** 

3. For historical reasons, the authors may consider citing a PhD thesis on the relationship between metabolic similarity and perceptual similarity by Chee-Ruiter (https://thesis.library.caltech.edu/7595/). I'm not sure if that work was ever published in a peer-reviewed journal.

This is a fantastic find and we are very appreciative of the reviewer for pointing this out to us. We have dedicated a full paragraph in the discussion to this wonderful work.

[Editors’ note: further revisions were suggested prior to acceptance, as described below.]

The manuscript has been substantially improved but there are some remaining issues that need to be addressed, as outlined below:1. Can the authors provide a more complete explanation of the analysis methods for Figure 4 supplement 1? There are very brief accounts in the methods section on pages 15 and 16. How does one go from ORN responses to regression targets and thence to neural distance?

We have added a more detailed explanation to the legend for Figure 4-supplement 1, starting with the p-values through the end of the legend.

2. The authors may wish to put some of the key points about Mordred vs POM measures from their responses to the reviews, into the text or discussion.

We have added these into the Results section starting with “Alternative structural distance metrics…” through the end of the paragraph.

3. Some fixes needed in the text:Page 12: "…in Figure 4 we computed distance metrics upon it is the distance between two components of an essential oil."

We have revised the sentence.

4. Figure 4 supplement 1: Carey et al. plot seems to be missing bars for cFP – edit distance.

The bars are there but are extremely small (close to zero). We have added the relevant values in the text above/below each bar.

5. Figure 4 supplement 1: Chae et al. seem to have a much smaller rank shift. Is this correct? Is it just due to the number of odors in the sample, or does it relate to the difference in the scoring function as described (very briefly) in the text?

Chae et al. has only 15 pairs of molecules (including repeated single molecules across pairs) that are also in the essential oil dataset, limiting its interpretability, i.e. analysis of Chae et al. is underpowered and noisy. We included it alongside the other two for completeness.

6. Figure 4 supplement 1: No stats to compare the bars?

We have added statistics to the legend.

7. Tanimoto distance should be spelled throughout with a capital T.

We have replaced the lowercase t with a capital T throughout.